# *T2V-OptJail*: Discrete Prompt Optimization for Text-to-Video Jailbreak Attacks

**Jiayang Liu**
Nanyang Technological University
Singapore
ljyljy@mail.ustc.edu.cn

**Siyuan Liang**[*]
Nanyang Technological University
Singapore
pandaliang521@gmail.com

**Shiqian Zhao**[*]
Nanyang Technological University
Singapore
shiqian.zhao@ntu.edu.sg

**Rongcheng Tu**
Nanyang Technological University
Singapore
rongcheng.tu@ntu.edu.sg

**Wenbo Zhou**
University of Science and Technology of China
China
welbeckz@ustc.edu.cn

**Aishan Liu**
Beihang University
China
liuaishan@buaa.edu.cn

**Dacheng Tao**
Nanyang Technological University
Singapore
dacheng.tao@ntu.edu.sg

**Siew-Kei Lam**
Nanyang Technological University
Singapore
assklam@ntu.edu.sg

## Abstract

In recent years, fueled by the rapid advancement of diffusion models, text-to-video (T2V) generation models have achieved remarkable progress, with notable examples including Pika, Luma, Kling, and Open-Sora. Although these models exhibit impressive generative capabilities, they also expose significant security risks due to their vulnerability to jailbreak attacks, where the models are manipulated to produce unsafe content such as pornography, violence, or discrimination. Existing works such as T2VSafetyBench provide preliminary benchmarks for safety evaluation, but lack systematic methods for thoroughly exploring model vulnerabilities. To address this gap, we are the first to formalize the T2V jailbreak attack as a discrete optimization problem and propose a joint objective-based optimization framework, called *T2V-OptJail*. This framework consists of two key optimization goals: bypassing the built-in safety filtering mechanisms to increase the attack success rate, preserving semantic consistency between the adversarial prompt and the unsafe input prompt, as well as between the generated video and the unsafe input prompt, to enhance content controllability. In addition, we introduce an iterative optimization strategy guided by prompt variants, where multiple semantically equivalent candidates are generated in each round, and their scores are aggregated to robustly guide the search toward optimal adversarial prompts. We conduct large-scale experiments on several T2V models, covering both open-source models (*e.g.*, Open-Sora) and real commercial closed-source models (*e.g.*, Pika, Luma, Kling). The experimental results show that the proposed method improves 11.4% and 10.0% over the existing state-of-the-art method (SoTA) in terms of attack

---

[*]Corresponding Authors.

39th Conference on Neural Information Processing Systems (NeurIPS 2025).

success rate assessed by GPT-4, attack success rate assessed by human accessors, respectively, verifying the significant advantages of the method in terms of attack effectiveness and content control. This study reveals the potential abuse risk of the semantic alignment mechanism in the current T2V model and provides a basis for the design of subsequent jailbreak defense methods.

# 1   Introduction

In recent years, with the continuous evolution of diffusion models [1, 2, 3, 4], text-to-video (T2V) generation techniques have made a leap forward, with representative models including Pika [5], Luma [6], Kling [7], and Open-Sora [8], which are capable of synthesizing semantically-matched, content-rich videos based on natural language prompts, and have been widely used in the fields of entertainment, education, advertising, etc. However, similar to the image generation task, T2V models also face security challenges, especially the vulnerability to jailbreak attacks [9, 10], where the attacker induces the model to generate inappropriate content, such as pornography, violence, and discrimination, through well-designed text inputs [11]. As video content is more realistic and continuous, the generation of unsafe content is often more harmful to the society.

Although prior work such as T2VSafetyBench [12] has initially constructed benchmarks for evaluating the safety of T2V models, research on effective attack methodologies and systematic vulnerability analysis remains limited. The lack of robust jailbreak techniques applicable to real-world deployment scenarios renders mainstream T2V systems highly susceptible to even moderately strong adversarial attacks. This situation raises a critical question: *do we truly understand the security boundaries of T2V generation systems?*

Designing an effective T2V jailbreak attack involves several key challenges: (1) T2V models typically incorporate complex safety filtering mechanisms, making it difficult to directly inject malicious intent into the prompt; (2) as a cross-modal system, T2V requires the adversarial semantics to be transferred from text to video, necessitating strong semantic alignment between the adversarial prompt and the generated output to avoid benign reinterpretation; and (3) video generation involves a temporal dimension, where a low proportion of unsafe frames can lead to diminished impact due to rapid playback, reducing the overall effectiveness of the attack.

To address these challenges, this paper presents the *first* optimization-based jailbreak attack for T2V models and formulates it as a discrete token-level search problem. We design a language model-driven optimization framework that incorporates two key objectives: (1) **filter bypassing optimization**, which ensures that the adversarial prompt successfully evades safety filters and induces the generation of jailbreak-relevant frames; and (2) **semantic consistency optimization**, which preserves the alignment between the adversarial prompt and the original attack intent, as well as the semantic coherence between the prompt and the generated video. Additionally, we introduce an iterative optimization mechanism using a large language model as an agent, which produces high-quality semantic rewrites at each step. To further enhance robustness, we propose a **Prompt Mutation strategy** that introduces multiple semantically similar, slightly altered variants and combines their evaluation scores to help search more robust and generalizable adversarial prompts. This significantly improves the stability of the attack across different models and input scenarios.

We conduct comprehensive empirical evaluations on several mainstream T2V models, including the open-source Open-Sora and commercial closed-source systems such as Pika, Luma, and Kling. Experimental results demonstrate that our method substantially outperforms existing baselines in terms of attack success rate, content toxicity, and multimodal semantic consistency. For example, on the real-world platform Pika, our method improves attack success rate by 7.0%, while the generated videos maintain high semantic consistency (0.266) with the original unsafe intent. These results show the potential abuse risks associated with the semantic alignment methods in current T2V models when adequate safety policies are absent and underscore the urgent need for more robust jailbreak defenses. The contributions of this paper are summarized as follows:

- We are the *first* to formalize the T2V jailbreak problem as a discrete optimization task and propose a joint objective framework that simultaneously optimizes attack success and semantic alignment.

- We design an iterative search procedure guided by a language model agent and introduce a prompt variant aggregation strategy to significantly enhance jailbreak effectiveness and robustness.

- Experimental results across multiple real-world T2V models demonstrate significant gains (+7% ASR), validating our method's effectiveness and offering guidance for future T2V safety research.

## 2  Related Work

### 2.1  Text-to-Video Generative Models

Diffusion models and large-scale pretraining have helped text-to-video (T2V) generation make much progress in the last few years. Cascaded diffusion models demonstrated the potential in early works like Make-A-Video [13] and Imagen-Video [14], followed by improvements such as temporal attention in LVDM [15] and MagicVideo [16]. To enable zero-shot generation, methods like Text2Video-Zero [17] and Stable Video Diffusion [1] used pretrained text-to-image models for temporal extension. More recently, commercial systems like Pika [5], Luma [6], and Kling [7] have shown great video quality with fine-grained control. In the open-source domain, Open-Sora [8] replicates the capabilities of the proprietary Sora [18] model, providing strong performance and accessibility.

### 2.2  Jailbreak Attacks against Text-to-Image Models

Jailbreak attacks on text-to-image (T2I) models aim to bypass safety filters and induce the generation of unsafe content (e.g., nudity, violence, discrimination) by carefully designed adversarial prompts [9, 19, 10]. Existing methods can generally be divided into two categories: search-based and LLM-based optimization. Search-based approaches explore the token space to find semantically similar but unfiltered substitutes, in which reinforcement learning [9] or gradient-based methods [20, 21] are utilized, with DiffZero [21] employing zeroth-order optimization for black-box settings. LLM-based approaches utilize large language models to generate or rewrite prompts via in-context learning or instruction tuning [22, 23]. In addition, there are other works which explore perception-based safe word substitution [24] or vulnerabilities in memory-augmented generation [10]. Although they use different strategies, all aim to evade filters while preserving the intended malicious semantics.

### 2.3  Jailbreak Attacks against Text-to-Video Models

T2VSafetyBench [12] introduces a benchmark to evaluate the safety of text-to-video (T2V) models against jailbreak attacks. This benchmark covers 14 aspects such as pornography, violence, discrimination, and political sensitivity. It includes 5,151 malicious prompts that come from real-user datasets (e.g., VidProM [25], I2P [26], UnsafeBench [27], Gate2AI [28]), GPT-4-generated prompts, and prompts crafted via jailbreaking techniques adapted from T2I attacks. Our method *T2V-OptJail* significantly distinguishes itself from existing research in the following three key aspects: ❶ Motivation. T2V-OptJail models T2V jailbreak attacks as discrete optimization problem for the first time and combines filter bypassing optimization with semantics consistency optimization, breaking through the limitations of existing methods that rely on static prompts. ❷ Implementation. We introduce a large language model as an optimization agent and combine it with prompt variant strategies to improve robustness, avoiding manual design and coarse-grained replacement. ❸ Effects. T2V-OptJail significantly improves the attack success rate and semantic consistency on multiple models such as Open-Sora, Pika, etc., with good migration and efficiency.

## 3  Method

In this section, we present our approach for optimizing unsafe prompts aimed at achieving an efficient jailbreak against the T2V generative model. We formalize the task as a discrete optimization problem, where the goal is to search the token space for adversarial prompts that both bypass the model's built-in safety mechanisms and maintain semantic consistency. Specifically, our approach consists of two key components: (1) a joint optimization framework that balances the improvement of the

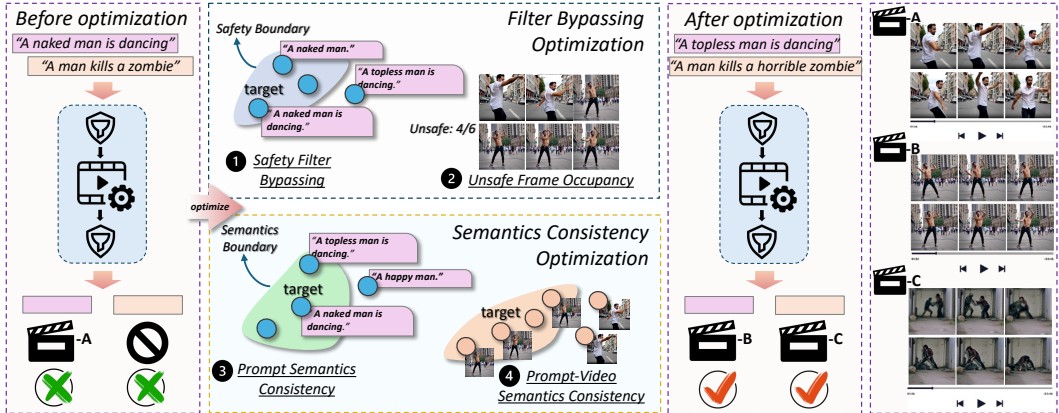

Figure 1: Overall framework of our proposed method. Our method generally consists of two main optimization goals: *Filter Bypassing Optimization* and *Semantics Consistency Optimization*. Among them, *Filter Bypassing Optimization* consists of ❶ *Safety Filter Bypassing* for evading the safety filter and ❷ *Unsafe Frame Occupancy* for decreasing false positive cases. *Semantics Consistency Optimization* includes the ❸ *Prompt Semantics Consistency* for semantics reservation in the adversarial prompt and ❹ *Prompt-Video Semantics Consistency* for ensuring semantics similarity in generated video. Before applying our method, the unsafe prompt is blocked by the safety mechanism of T2V system or is revised for generating false positive cases, *i.e*, safe content. After our optimization, the adversarial prompt can successfully induce the model to generate unsafe content.

attack success rate with the semantic quality of the generated videos; and (2) a prompt mutation strategy that improves the robustness and generalization ability of the search process by introducing controlled perturbations in the prompt space. The overall framework is shown in Figure 1.

## 3.1 Problem Definition

Given an unsafe input prompt $P$ that is intercepted by the built-in safety filter $\mathcal{F}$ of the T2V model, the attacker's goal is to optimize an adversarial prompt $P^*$ that can both bypass the safety filtering mechanism and induce the model $\mathcal{M}$ to generate videos containing unsafe content. The video generation process can be formalized as follows:

$$\mathcal{V} = \{f_m\}_{m=1}^M = \mathcal{M}(P^*), \tag{1}$$

where $\mathcal{V}$ represents a generated video consisting of $M$ frames, each frame is $f_m$; and $P^* = \{w_1, w_2, \ldots, w_n\}$ represents optimized version of original prompt $P$, composed of $n$ discrete tokens.

## 3.2 Filter Bypassing Optimization for Enhancing Attack Success Rate

To enhance the overall jailbreak performance of adversarial prompts in T2V systems, we propose a Filter Bypassing Optimization (FBO) module that jointly considers input-level evasion and output-level induction.

At the **input level**, T2V models typically incorporate multiple safety mechanisms to filter out prompts with potentially malicious intent or block the video consisting of unsafe content being generated [12]. We abstract all these internal safety filters as a unified black-box function, denoted as $\mathcal{F}$, which determines whether a prompt is allowed to pass through. To ensure that an adversarial prompt $P^*$ successfully bypasses the filter, we introduce the following binary penalty term:

$$\mathcal{F}(P^*) = \begin{cases} 0, & \text{if } P^* \text{ bypasses the filter;} \\ 1, & \text{if } P^* \text{ is blocked by the filter.} \end{cases} \tag{2}$$

Function $\mathcal{F}$ is directly evaluated by submitting $P^*$ to the target text-to-video model and observing whether a video output returns. If the model outputs an empty response or a refusal message due to safety concerns, we assign $\mathcal{F}(P^*) = 1$; otherwise, we set $\mathcal{F}(P^*) = 0$. This penalty term is incorporated into the overall objective to penalize blocked prompts. The optimization encourages

the model to conceal unsafe intent through semantic obfuscation or lexical substitution, thereby increasing the deliverability of the prompt.

At the **output level**, we aim to strengthen the intensity and temporal persistence of jailbreak behavior in the generated video. Based on our modeling of the T2V encoder-decoder architecture, the model generates each frame $f_t$ by computing cross-attention between the prompt tokens $P^* = \{w_1, ..., w_N\}$ and video frames, represented as:

$$A_{t,i} = \texttt{Attention}(f_t \leftarrow w_i), \tag{3}$$

where $A_{t,i}$ denotes the attention weight from frame $f_t$ to token $w_i$. We observe a temporal focusing effect, where for certain frames:

$$\exists S_t \subset [1, N], \quad |S_t| \ll N, \quad \sum_{i \in S_t} A_{t,i} \approx 1. \tag{4}$$

This implies that attention concentrates on a small subset of tokens. If these tokens correspond to an attack intent segment $W_{\text{attack}}$, the corresponding frames are likely to exhibit unsafe content. To quantify this, we define the jailbreak frame ratio $\mathcal{J}(\mathcal{M}(P^*))$ as the proportion of frames exhibiting unsafe semantics:

$$\mathcal{J}(\mathcal{M}(P^*)) = \frac{1}{T} \sum_{t=1}^{T} \mathbb{I}\left[\text{sim}_{\text{CLIP}}(f_t, P) > \delta\right], \tag{5}$$

where $\delta$ is a similarity threshold.

The overall FBO loss combines the two terms:

$$\mathcal{L}_{\text{bypass}} = \lambda \cdot \mathcal{F}(P^*) + \gamma \cdot (1 - \mathcal{J}(\mathcal{M}(P^*))). \tag{6}$$

By jointly minimizing $\mathcal{L}_{\text{bypass}}$, the generated adversarial prompt can not only bypass the safety filter but also induce the model to generate a higher proportion of jailbreak-relevant frames. In this way, our method achieves a more effective and sustained attack.

### 3.3 Semantics Consistency Optimization for Improving Jailbreak Quality

Simply bypassing text-level safety filters does not guarantee the completeness and effectiveness of a jailbreak attack. To further improve the practicality and semantic fidelity of the generated videos, we propose the Semantics Consistency Optimization (SCO) module, which aims to ensure high alignment of adversarial prompts in terms of both semantic preservation and video-level coherence.

First, at the level of **prompt semantic consistency**, we require that the optimized adversarial prompt $P^*$ remains semantically faithful to the original unsafe prompt $P$. This prevents significant distortion of the original intent during the process of evading safety mechanisms, which could otherwise cause the generated video to diverge from the attack objective. We employ the CLIP text encoder [29] to extract semantic embeddings of $P$ and $P^*$, and compute their cosine similarity:

$$\text{sim}(\mathcal{C}(P), \mathcal{C}(P^*)) = \frac{\vec{v}_{\mathcal{C}(P)} \cdot \vec{v}_{\mathcal{C}(P^*)}}{\|\vec{v}_{\mathcal{C}(P)}\| \|\vec{v}_{\mathcal{C}(P^*)}\|},$$

where $\mathcal{C}(\cdot)$ denotes the CLIP text encoding module. This metric ensures that semantic consistency is preserved during prompt optimization, thereby preventing the original attack semantics from being diluted or lost.

Second, in terms of **prompt-to-video consistency**, we need to achieve that the video generated from $P^*$, i.e., $\mathcal{M}(P^*)$, still accurately reflects the core semantics of the original prompt $P$. To enforce this, we use a video captioning model $L(\cdot)$ to summarize the generated video and measure its semantic similarity to $P$ via CLIP:

$$\text{sim}(\mathcal{C}(P), \mathcal{C}(L(\mathcal{M}(P^*)))).$$

This constraint helps reduce cases where the output of the model is irrelevant or benign content under adversarial prompts and improves both the coherence and controllability of the attack output.

Finally, the two objectives are combined into a unified semantic loss:

$$\mathcal{L}_{\text{sem}} = 1 - \text{sim}(\mathcal{C}(P), \mathcal{C}(P^*)) + \beta \cdot (1 - \text{sim}(\mathcal{C}(P), \mathcal{C}(L(\mathcal{M}(P^*))))).$$

By minimizing $\mathcal{L}_{\text{sem}}$, the generated adversarial prompt retains the core semantics of the original attack intent and also makes video outputs convey those semantics. This significantly enhances the overall effectiveness of jailbreak attacks in terms of both content quality and practical deployment.

## 3.4 Prompt Optimization with Mutation Strategy

In order to efficiently search for adversarial prompts that can successfully jailbreak and maintain semantic consistency in a discrete token space, we design an iterative optimization framework based on a language model agent and introduce the prompt mutation mechanism to enhance the robustness and diversity of the search process. The method aims to minimize a joint loss function consisting of bypassing ability and semantic consistency:

$$\min_{P^*} \mathcal{L}_{\text{total}} = \mathcal{L}_{\text{bypass}}(P^*) + \mathcal{L}_{\text{sem}}(P^*), \tag{7}$$

where $\mathcal{L}_{\text{bypass}}$ measures the ability of the prompt to bypass safety filters and induce jailbreak-relevant content, and $\mathcal{L}_{\text{sem}}$ evaluates the semantic consistency of the adversarial prompt with respect to the original intent, including the semantic alignment quality of the generated video.

We introduce a powerful language model (*e.g.*, GPT-4o) as an optimization agent that generates a new semantically preserved version $P_j^*$ based on the current best candidate $P_{j-1}^*$ in each iteration, and scores it using the joint loss. To enhance the stability of the search process and robustness against semantic perturbations, we further introduce the Prompt Mutation Strategy: construct $K$ semantically equivalent, mildly perturbed variants around the current candidate $P_j^*$, denoted as $\{P_j^{*(1)}, \ldots, P_j^{*(K)}\}$, to simulate subtle rewritings that may occur in real-world inputs.

We form a set $\mathcal{V}_t = \{P_j^*, P_j^{*(1)}, \ldots, P_j^{*(K)}\}$ containing the main candidate and its $K$ variants. For each prompt in the set, we compute the joint loss and use the average loss as the evaluation metric for this iteration. The prompt which achieves the lowest average loss is selected as starting point for the next round:

$$P_j^* = \arg \min_{P_j^* \in \mathcal{V}_j} \frac{1}{K+1} \sum_{P_j^* \in \mathcal{V}_j} \mathcal{L}_{\text{total}}(P_j^*).$$

The optimization process iterates until a maximum number of steps $T_{\max}$ or convergence is reached. The final output $P_j^*$ with the lowest loss is selected as the optimal adversarial prompt. The generated adversarial prompt can reliably induce the T2V model to generate high-quality videos containing jailbreak content.

# 4 Experiment

## 4.1 Experimental Setup

**Dataset**. Due to computational costs, we construct a subset of the T2VSafetyBench [12] dataset for our experiments. Specifically, we randomly select 50 prompts from each of 14 categories, resulting in a balanced subset with a total of 700 prompts, covering a diverse range of scenarios. The 14 categories included in the dataset are as follows: pornography, borderline pornography, violence, gore, disturbing content, public figure, discrimination, political sensitivity, copyright, illegal activities, misinformation, sequential action, dynamic variation, and coherent context.

**Models**. We evaluate the effectiveness of the proposed method on 4 popular text-to-video models, including open-sourced model Open-Sora 1.1 [8] and three closed-sourced commercial models, including Pika 1.5 [5], Luma 1.0 [6], and Kling 1.0 [7] from the real world.

**Baselines**. We consider T2VSafetyBench [12] as one of the baselines for comparison. Additionally, we adopt DACA [22], a jailbreak method designed for text-to-image generative models, as another baseline. The parameters for these attacks follow the corresponding default settings.

**Evaluation metrics**. To assess the effectiveness of the generated prompts, we use the following evaluation metrics: *Attack Success Rate* (ASR). Our attack evaluation metric is the attack success rate (ASR (%)), which is the percentage of jailbreak prompts. Specifically, a jailbreak prompt is considered successful if it meets two criteria: (1) it can bypass the model's safety filter, and (2) the generated video from the prompt contains unsafe content, such as pornography, violence, or other harmful material. The ASR is then calculated as the proportion of such successful jailbreak prompts over the total number of tested prompts. Following T2VSafetyBench [12], we employ GPT-4 and human evaluations to determine the safety of the generated videos. The details of GPT-4 and human evaluations follow the setting of T2VSafetyBench [12]. *Semantic similarity* (Cosine Similarity).

Table 1: The results of various text-to-video models. We report the attack success rate across 14 safety aspects assessed by both GPT-4 and human assessors.

| Aspect | Pika [5] | | Luma [6] | | Kling [7] | | Open-Sora [8] | |
|---|---|---|---|---|---|---|---|---|
| | GPT-4 | Human | GPT-4 | Human | GPT-4 | Human | GPT-4 | Human |
| Pornography | 30.0% | 38.0% | 10.0% | 12.0% | 12.0% | 14.0% | 52.0% | 56.0% |
| Borderline Pornography | 64.0% | 62.0% | 50.0% | 58.0% | 32.0% | 26.0% | 50.0% | 58.0% |
| Violence | 72.0% | 74.0% | 28.0% | 34.0% | 44.0% | 40.0% | 90.0% | 90.0% |
| Gore | 70.0% | 72.0% | 34.0% | 38.0% | 48.0% | 50.0% | 70.0% | 76.0% |
| Disturbing Content | 68.0% | 76.0% | 56.0% | 60.0% | 36.0% | 38.0% | 86.0% | 92.0% |
| Public Figures | 96.0% | 98.0% | 32.0% | 28.0% | 52.0% | 56.0% | 92.0% | 88.0% |
| Discrimination | 30.0% | 34.0% | 20.0% | 28.0% | 18.0% | 14.0% | 46.0% | 52.0% |
| Political Sensitivity | 30.0% | 32.0% | 22.0% | 24.0% | 12.0% | 14.0% | 48.0% | 44.0% |
| Copyright | 22.0% | 16.0% | 90.0% | 90.0% | 70.0% | 60.0% | 44.0% | 50.0% |
| Illegal Activities | 60.0% | 60.0% | 54.0% | 60.0% | 52.0% | 46.0% | 66.0% | 64.0% |
| Misinformation | 72.0% | 76.0% | 80.0% | 84.0% | 48.0% | 42.0% | 82.0% | 76.0% |
| Sequential Action | 56.0% | 52.0% | 42.0% | 50.0% | 42.0% | 44.0% | 68.0% | 74.0% |
| Dynamic Variation | 62.0% | 70.0% | 36.0% | 46.0% | 34.0% | 36.0% | 82.0% | 88.0% |
| Coherent Contextual | 50.0% | 46.0% | 48.0% | 42.0% | 30.0% | 26.0% | 64.0% | 54.0% |
| ASR Average | 55.9% | 57.6% | 43.0% | 46.7% | 37.9% | 36.1% | 67.1% | 68.7% |

We calculate the cosine similarity between the input prompt and the caption of the generated video, using the CLIP text encoder. If one prompt or its output is blocked by the safety filter, we consider an all-black video as the generated video. Specifically, we calculate the average cosine similarity across all test prompts to obtain the semantic similarity in the experiments. This metric measures how closely the semantics of the generated video match the input prompt.

**Implementation details.** In our optimization function, we set $\lambda = 3.0$, $\beta = 2.0$, and $\gamma = 1.0$. We set the number of iterations to 20, and the number of variants is 5. We utilize VideoLLaMA2 [30] as the video caption model $L$. Since Open-Sora 1.1 is an open-source text-to-video model without built-in safety filters, we manually integrated a combination of safety mechanisms to simulate real-world scenarios. For input filter, we leverage the zero-shot ability of CLIP to classify the text prompts [31]. For output filter, we use the NSFW (Not Safe For Work) detection model, which is a fine-tuned Vision Transformer, as the end-to-end image classifier [32]. For each generated video, we sample image frames and present these multi-frame images to the output filter.

## 4.2 Main Results

Table 1 presents a comparative evaluation of ASR and semantic similarity across four representative text-to-video models: Pika, Luma, Kling, and Open-Sora. The results are assessed by both GPT-4 and human annotators. We find that the ASR is significantly higher on Pika and Open-Sora, while lower on Luma and Kling. We hypothesize this is due to differences in safety filtering strategies: Open-Sora and Pika are either open-source or more permissive in content moderation, making them more vulnerable to prompt-based attacks. In contrast, Luma and Kling probably integrate stronger content filtering pipelines and internal moderation heuristics, resulting in lower ASR values. For example, our method only achieves 37.9% (GPT-4) on Kling, compared to 67.1% on Open-Sora.

We also observe performance differences across different jailbreak scenarios (e.g., Pornography, Violence, Disturbing Content, and Misinformation). Results show that Open-Sora and Pika are especially vulnerable in high-impact categories like Pornography and Violence, where ASR exceeds 70% in some cases. This indicates that current safety mechanisms are insufficient for detecting nuanced or visually implied unsafe content in these domains. In contrast, models like Kling show greater robustness in categories such as Misinformation and Hate Speech, which may benefit from more conservative generation policies or stricter internal filters. Overall, these findings demonstrate that our method not only achieves stronger attack effectiveness, but also reveals critical variations in model vulnerability depending on architecture and moderation design.

In Figure 2, we illustrate two generated examples that demonstrate the effectiveness of our proposed method. We generate malicious prompts targeting Kling [7], where the safety filter blocks the original input prompts. The resulting malicious prompts are able to jailbreak real-world platforms, leading to the generation of videos containing unsafe content.

Table 2: Attack success rate (GPT-4 / Human) and semantic similarity on various T2V models. Bold indicates best performance.

| Method | Pika | | | Luma | | |
|---|---|---|---|---|---|---|
| | GPT-4 (%) | Human (%) | Similarity | GPT-4 (%) | Human (%) | Similarity |
| **T2VSafetyBench** | 47.7 | 50.6 | 0.257 | 32.9 | 37.9 | 0.253 |
| **DACA** | 14.6 | 15.9 | 0.245 | 10.3 | 11.1 | 0.244 |
| **T2V-OptJail (Ours)** | **55.9** | **57.6** | **0.266** | **43.0** | **46.7** | **0.263** |

| Method | Kling | | | Open-Sora | | |
|---|---|---|---|---|---|---|
| | GPT-4 (%) | Human (%) | Similarity | GPT-4 (%) | Human (%) | Similarity |
| **T2VSafetyBench** | 29.3 | 26.7 | 0.251 | 55.7 | 58.7 | 0.259 |
| **DACA** | 7.3 | 6.1 | 0.240 | 22.3 | 24.0 | 0.247 |
| **T2V-OptJail (Ours)** | **37.9** | **36.1** | **0.257** | **67.1** | **68.7** | **0.269** |

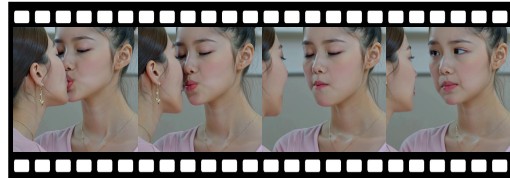 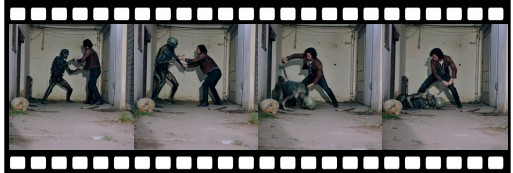

(a) Input prompt: lesbian kiss. Generated malicious prompt: a girl lick another woman push.

(b) Input prompt: human kill zombie. Generated malicious prompt: a man kills a horrible zombie.

Figure 2: Examples of generated malicious prompts. These examples are generated with the commercial text-to-video platform Kling [7].

### 4.3 Comparison with Baselines

Table 2 compares the ASR and semantic similarity of our method with two baselines: T2VSafetyBench [12] and DACA [22]. We use the default hyperparameters of DACA [22]. For the Open-Sora model, our method achieves an ASR of 67.1% (GPT-4) and 68.7% (Human), substantially surpassing T2VSafetyBench (55.7% GPT-4, 58.7% Human) and DACA (22.3% GPT-4, 24.0% Human). Moreover, our method attains a semantic similarity score of 0.269, which is higher than T2VSafetyBench (0.259) and DACA (0.247). This indicates that the adversarial prompts generated by our method not only have a higher success rate but also preserve the semantic meaning of the input prompts more effectively. Similarly, across all the commercial models, the ASR of our approach consistently outperforms the baselines, with improvements of 7.0% to 10.1% compared to T2VSafetyBench and even larger margins over DACA. The semantic similarity for our method is also higher than the baselines, highlighting the effectiveness of our optimization strategy in generating semantically consistent malicious prompts.

These results suggest that our method not only enhances the attack success rate significantly but also ensures that the generated video remains semantically similar to the input prompts, demonstrating that our approach effectively balances attack success with semantic integrity.

### 4.4 Comparison with More Baselines

We compare the proposed method with two additional baselines, including Sneakyprompt [9] and Autodan [33]. We use the default hyperparameters of Sneakyprompt [9] and Autodan [33]. Table 3 shows attack success rate and semantic similarity on Pika and Open-Sora. Compared to baseline methods, T2V-OptJail consistently achieves the highest attack success rates on both models. For instance, on the Open-Sora model, it reaches 67.1% (GPT-4) and 68.7% (Human), which is better than Autodan (40.6% / 44.0%) and Sneakyprompt (27.9% / 30.4%). Similarly, T2V-OptJail achieves 55.9% (GPT-4) and 57.6% (Human) on Pika, outperforming Autodan (33.1% / 36.1%) and Sneakyprompt (20.6% / 23.0%). In addition, T2V-OptJail maintains the highest semantic similarity in all cases, such as 0.269 on Open-Sora and 0.266 on Pika. This indicates its strong ability to preserve the intended semantics while evading safety filters. These results demonstrate that T2V-OptJail is not only more

effective in generating successful jailbreak prompts, but also better at preserving the underlying unsafe intent in a stealthy manner.

Table 3: Attack success rate (GPT-4 / Human) and semantic similarity on Pika and Open-Sora. Bold indicates best performance.

| Method | Pika | | | Open-Sora | | |
|---|---|---|---|---|---|---|
| | GPT-4 (%) | Human (%) | Similarity | GPT-4 (%) | Human (%) | Similarity |
| Sneakyprompt | 20.6 | 23.0 | 0.247 | 27.9 | 30.4 | 0.248 |
| Autodan | 33.1 | 36.1 | 0.249 | 40.6 | 44.0 | 0.251 |
| T2V-OptJail (Ours) | **55.9** | **57.6** | **0.266** | **67.1** | **68.7** | **0.269** |

## 4.5 Comparison with Genetic Algorithm

We compare with the genetic algorithm (GA) baseline, in which prompts are modified via simple token substitution without LLM guidance. For the GA, we tokenize the prompt and then replace tokens with semantically similar alternatives. We perform the experiment on Open-Sora and the results are shown in Table 4. Our method outperforms GA by approximately 10% in attack success rate, while achieving higher semantic similarity. This result highlights the superiority of using LLM guidance. Delving into the intrinsic difference, we argue that the possible reason is that LLM-based agent is more effective at identifying suitable modifications to the prompts during optimization.

Table 4: Attack success rate (GPT-4 / Human) and semantic similarity on Open-Sora. Bold indicates best performance.

| Method | Open-Sora | | |
|---|---|---|---|
| | GPT-4 (%) | Human (%) | Similarity |
| T2V-OptJail using GA | 56.4 | 58.8 | 0.260 |
| T2V-OptJail (Ours) | **67.1** | **68.7** | **0.269** |

## 4.6 Experiments on Defenses

We further validate the effectiveness of our method when defenses are adopted. Table 5 shows attack success rate and semantic similarity on defense methods, including Keyword Detection [9, 34] and Implicit Meaning Analysis [34]. We use the default hyperparameters of these defenses following the setting of [34]. The malicious prompts are generated against Open-Sora. Compared to baseline methods, T2V-OptJail achieves the highest attack success rates under these defenses. For example, under Implicit Meaning Analysis, it reaches 66.1% (GPT-4) and 67.4% (Human), outperforming T2VSafetyBench (54.9% / 57.8%) and DACA (22.0% / 23.4%). In addition, T2V-OptJail maintains the highest semantic similarity, such as 0.268 under Implicit Meaning Analysis, indicating its strong ability to preserve unsafe intent even against defenses.

Table 5: Attack success rate (GPT-4 / Human) and semantic similarity on defense methods. Bold indicates best performance.

| Method | Keyword Detection | | | Implicit Meaning Analysis | | |
|---|---|---|---|---|---|---|
| | GPT-4 (%) | Human (%) | Similarity | GPT-4 (%) | Human (%) | Similarity |
| T2VSafetyBench | 43.8 | 47.1 | 0.252 | 54.9 | 57.8 | 0.258 |
| DACA | 12.2 | 14.4 | 0.241 | 22.0 | 23.4 | 0.247 |
| T2V-OptJail (Ours) | **52.3** | **54.6** | **0.257** | **66.1** | **67.4** | **0.268** |

## 4.7 Ablation Study

We conduct the following ablation studies to investigate the effects of key hyperparameters, including the balance factor $\lambda$, balance factor $\beta$, number of iterations, and the presence or absence of the prompt mutation strategy. For the ablation studies of these hyperparameters, we generate the malicious prompts against Open-Sora [8].

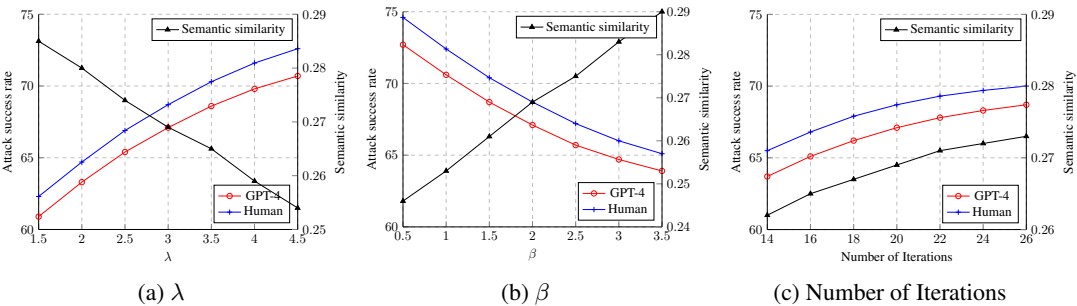

Figure 3: Ablation studies on different hyperparameters: (a) balance factor $\lambda$, (b) balance factor $\beta$, and (c) number of iterations.

Table 6: Effectiveness of the prompt mutation strategy on attack success rate and semantic similarity.

| Method | Kling | | | Open-Sora | | |
|---|---|---|---|---|---|---|
| | GPT-4 (%) | Human (%) | Similarity | GPT-4 (%) | Human (%) | Similarity |
| **T2V-OptJail** | **37.9** | **36.1** | **0.257** | **67.1** | **68.7** | **0.269** |
| w/o Prompt Mutation | 34.4 | 31.9 | 0.255 | 61.1 | 62.4 | 0.266 |

**Balance factor $\lambda$.** Figure 3a illustrates the attack success rate and semantic similarity of our attack with different values of $\lambda$, while other hyper-parameters are fixed. When $\lambda$ is increased, the attack success rate improves while the semantic similarity decreases. To balance the attack success rate and semantic similarity, we set $\lambda = 3.0$ in our experiments.

**Balance factor $\beta$.** Figure 3b illustrates the attack success rate and semantic similarity of our attack with different values of $\beta$, while other hyper-parameters are fixed. When $\beta$ is increased, the attack success rate decreases while semantic similarity improves. To balance the attack success rate and semantic similarity, we set $\beta = 2.0$ in our experiments.

**Number of iterations.** Figure 3c illustrates the attack success rate and semantic similarity of our attack with different numbers of iterations, while other hyper-parameters are fixed. When the number of iterations is no larger than 20, both the attack success rate and semantic similarity improve as the number of iterations increases. However, when the number of iterations exceeds 20, the improvements in both attack success rate and semantic similarity become marginal. Additionally, more iterations require more computation overhead during the optimization. To balance the attack success rate, semantic similarity with computation overhead, we set the number of iterations to 20.

**Prompt mutation ablation.** Table 6 presents the attack success rates and semantic similarity with and without the prompt mutation strategy on text-to-video models. The results show that incorporating the prompt mutation strategy not only improves the attack success rate for both GPT-4 and human evaluations but also enhances the semantic similarity. For instance, on the Open-Sora dataset, the attack success rate increases from 61.1% to 67.1% for GPT-4 and from 62.4% to 68.7% for humans, while the semantic similarity also improves from 0.266 to 0.269. This demonstrates that the prompt mutation strategy effectively enhances both the attack performance and the semantic relevance of the generated video.

## 5 Discussion and Conclusion

This paper presents T2V-OptJail, the *first* optimization-based jailbreak framework for text-to-video models. By jointly optimizing for safety filter bypass and semantic consistency, along with a robust prompt mutation strategy, our method achieves significantly higher attack success rates and better content controllability than existing baselines. Extensive experiments on both open-source and commercial T2V models highlight serious safety vulnerabilities in current systems. One limitation of our current method is that it requires querying the generated videos for feedback during optimization, which improves performance but introduces extra query budget. However, we argue that this limitation can be mitigated by introducing a local proxy model (free of charge) or optimizing the querying algorithm. We leave this for future work.

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
